# Treatment Outcome in Patients with *Mycobacterium abscessus* Complex Lung Disease: The Impact of Tigecycline and Amikacin

**DOI:** 10.3390/antibiotics11050571

**Published:** 2022-04-25

**Authors:** Jeng-How Yang, Ping-Huai Wang, Sheng-Wei Pan, Yu-Feng Wei, Chung-Yu Chen, Ho-Sheng Lee, Chin-Chung Shu, Ting-Shu Wu

**Affiliations:** 1Division of Infectious Diseases, Department of Internal Medicine, Chang Gung Memorial Hospital, Linkou Medical Center, Chang Gung University College of Medicine, No. 5, Fuxing St., Guishan Dist., Taoyuan City 33302, Taiwan; summerfield8731113@gmail.com; 2Department of Internal Medicine, Far-Eastern Memorial Hospital, New Taipei City 30010, Taiwan; pinghuaiwang@gmail.com; 3School of Medicine, National Yang-Ming University, Taipei 22000, Taiwan; swpan25@gmail.com; 4Department of Chest Medicine, Taipei Veterans General Hospital, Taipei 11267, Taiwan; 5Department of Internal Medicine, E-Da Hospital, Kaohsiung City 82445, Taiwan; yufeng528@gmail.com (Y.-F.W.); leehoshn@gmail.com (H.-S.L.); 6Department of Internal Medicine, National Taiwan University Hospital Yunlin Branch, Yunlin County 64041, Taiwan; y000676@gmail.com; 7College of Medicine, National Taiwan University, Taipei 10607, Taiwan; 8Department of Internal Medicine, National Taiwan University Hospital, No. 7, Chung Shan South Road, Taipei 10002, Taiwan; 9School of Medicine, Chang Gung University, No. 259, Wenhua 1st Rd., Guishan Dist., Taoyuan City 33302, Taiwan

**Keywords:** *Mycobacterium abscessus* complex, lung disease, tigecycline, amikacin, treatment outcome

## Abstract

Background: The contemporary guidelines have recommended multiple antimicrobial therapies along with oral macrolides for the treatment of *Mycobacterium abscessus* complex lung disease (MABC-LD). However, there is little evidence supporting the parenteral tigecycline-containing regimens against MABC-LD. Therefore, we conducted this study to evaluate the effect of intravenous tigecycline-containing regimens on the treatment of MABC-LD. Methods: A retrospective study was conducted in 6 medical centers. Patients with MABC-LD that were followed up at ≥12 months were enrolled. *Mycobacterium abscessus* subspecies were identified by *hsp65*, *rpoB*, *secA1* gene PCR, and sequencing. Antimicrobial susceptibility was determined for 34 patients using broth microdilution methods following the Clinical and Laboratory Standards Institute (CLSI) guideline. The microbiology and treatment outcomes were defined as either success or failure. The impacts of tigecycline and amikacin were adjusted for age, comorbidities, surgical resection, and radiologic scores. Results: During the study period, seventy-one patients were enrolled for final analysis. The microbiology failure rate was 61% (43/71) and the treatment failure rate was 62% (44/71). For *M. abscessus* complex, 97% (33/34) of tigecycline MIC were ≤1 mg/L. Amikacin also demonstrated great susceptibility (94.1%; 32/34). Treatment with regimens containing tigecycline plus amikacin provided better microbiology success (adjusted OR 17.724; 95% CI 1.227–267.206) and treatment success (adjusted OR 14.085; 95% CI 1.103–166.667). Conclusion: The outcome of MABC-LD is always unsatisfactory. Treatment regimens with oral macrolide in combination with tigecycline and amikacin were correlated with increased microbiology success and less treatment failure.

## 1. Introduction

Nontuberculous mycobacterial lung disease (NTM-LD) has become an important concern [1] because its prevalence and incidence have increased over the recent two decades [2,3]. The increasing trend is reported not only in Taiwan [4] but also in the US [5], Korea [6], and Japan [7]. In addition to the increasing burden, the poor long-term prognosis of NTM-LD calls for focused attention. The overall mortality rates in NTM-LD patients at 5 and 10 years were 12.4 and 24.0% [8]. Among the different NTM species, *Mycobacterium abscessus* complex (MABC) has a higher hazard ratio (2.19) of mortality than that of *M. avium* complex (MAC) [8]. Many factors contribute to the high mortality in MABC lung disease (MABC-LD), and among them, the poor treatment success rate might be the most important. A recent meta-analysis reported a treatment success rate of only 41.2% for MABC-LD [9]. The high resistance rate of MABC and lack of effective drugs [10] might explain the low treatment success rate of MABC-LD.

In the contemporary treatment guideline for MABC-LD [11,12], around 1–3 months of intravenous drugs together with effective oral antibiotics, especially new macrolides, are recommended for the intensive treatment phase for MABC-LD. However, patient compliance is not good. More than half of the patients may not receive intravenous agents [13], possibly due to the patient’s fear, severe adverse drug effects and scarcity of evidence supporting the treatment. In a previous individual-data meta-analysis, intravenous (IMI) and amikacin (AMI) were shown to improve treatment success [14]. Tigecycline (TGC), a new intravenous agent for MABC [15], has yet to be studied well for MABC-LD. Although previous studies have shown that MABC has high in vitro susceptibility to TGC [16], the microbiological responses of MABC-LD conflicted in previous reports [17,18]. In particular, a direct comparison between regimens with and regimens without TGC is needed. Therefore, we conducted this study to investigate the effect of an intravenous TGC containing regimen on the treatment of MABC-LD.

## 2. Methods

### 2.1. Study Population

This retrospective study was conducted with a medical chart review and bacterial analysis of MABC-LD patients. The patient population was enrolled from 6 medical centers in Taiwan, including National Taiwan University Hospital (NTUH, Taipei City, Taiwan) from 2011 to 2017; NTUH, YunLin (YL) branch, from 2014 to 2017; Linkou Chang Gung Memorial Hospital (CGMH, Taoyuan, Taiwan) from 2017 to 2020; Taipei Veterans General Hospital (TVGH, Taipei, Taiwan) from 2014 to 2017; E-Da Hospital (EDH, Kaohsiung, Taiwan) from 2011 to 2017; and the Far Eastern Memorial Hospital (FEMH, New Taipei City, Taiwan) from 2009 to 2018. Patients meeting MABC-LD diagnosis criteria recommended by the American Thoracic Society (ATS)/Infectious Diseases Society of America (IDSA) and receiving treatment were enrolled. Patients aged under 20 years, with a follow-up period of less than 12 months and infection with TB or other NTM at the same time were excluded from the study. This study was approved by the Institutional Review Board of the Research Ethics Committees of all hospitals (NTUH and its YL branch: 201704001RINB, CGMH: 201601809B0, TVGH: 2017-09-010C, EDH: EMRP-108-066, and FEMH: 107162-E).

### 2.2. Mycobacterium Abscessus Complex Clinical Strains

Clinical MABC isolates were identified as *M. abscessus* subsp. *abscessus* (MAB), or *M. abscessus* subsp. *massiliense* (MMA) for strains in TVGH, CGMH and NTUH retrospectively. The subspecies identification was based on gene sequences of *hsp65*, *rpoB*, and *secA1* genes, which were examined using polymerase chain reaction (PCR) and then by Sanger sequencing. The details of PCR sequencing were described by Zelazny et al. [19]. Genetic information of *hsp65*, *rpoB*, *secA1* were compared with the database from the NCBI (http://blast.ncbi.nlm.nih.gov/Blast.cgi (accessed on 28 September 2020)).

### 2.3. Antimicrobial Susceptibility Testing (AST)

The AST was determined by broth microdilution method (BMD) for available strains in CGMH and NTUH, retrospectively. Serial dilutions of tested drugs were prepared with Sensititre^TM^ RAPMYCOI^TM^ MIC plates (Thermo Fisher, Waltham, MA, USA). Eleven antimicrobial agents were tested for the minimal inhibitory concentration, as follows: AMI, cefoxitin (FOX), ciprofloxacin (CIP), clarithromycin (CLA), doxycycline (DOX), IMI, linezolid (LZD), minocycline (MIN), moxifloxacin (MXF), TGC, and trimethoprim/sulfamethoxazole (SXT). The breakpoints of minimal inhibitory concentrations (MICs) were determined by the CLSI recommendations [20] except TGC [16]. The MICs of CLA were determined at an early reading time (ERT; usually from the 3rd to 5th day) and a delayed reading time (LRT; the 14th day of incubation) for detecting the inducible macrolide resistance.

### 2.4. Administration and Assessment of Effective Treatment Regimens

The assessment of effective treatments for MABC-LD was based on 2020 ATS/IDSA official statements [12]. Periodic administration of multiple (≥two kinds) effective ant-MABC agents was considered an effective regimen. Parenteral agents including AMI, cefoxitin, IMI, fluoroquinolone (FQ) or TGC ≥ 4 weeks were recorded as effective. A peripherally inserted central catheter was inserted if prolonged intravenous antibiotics was administered. The dosing of intravenous anti-MABC agents were based on ATS/IDSA guideline [12]: AMI, 8–10 mg/kg for 3 to 5 days per week; TGC, 100 mg on the initial day 1 and then 50 mg per day; IMI, 250 mg every six hours; moxifloxacin, 400 mg daily; and levofloxacin, daily 750 mg. The oral dose of azithromycin was 500 mg on day 1, followed by 250 mg per day, and clarithromycin was 500 mg twice daily. The use of anti-MABC agents was according to clinical response, and the above-mentioned subspecies identification and antimicrobial susceptibility testing were not obtained for the clinician.

### 2.5. Chest Radiographical Scoring

Initial posterior–anterior chest X-rays were obtained for each patient. The patterns of radiographical lesions were classified as nodular bronchiectasis and fibro-cavitation change. Radiographical scores were measured as in a previous report [21]. In brief, the lung field was divided into 3 zones according to two horizontal lines located at the distal end of the lobar pulmonary artery. Each zone was rated 0 to 3 points according to lesions involving less than 1/3 (1 point), 1/3–2/3 (2 points), or more than 2/3 (3 points) of the zone field [21]. A sum of scores over six zones is the score, which ranges from 0 to 18.

### 2.6. Assessment of Outcomes

The outcomes were classified as a microbiology cure and treatment success after a 12-month period follow-up. The definition of microbiology cure was finding multiple consecutive negative cultures for causative MABC from respiratory samples after culture conversion and until the end of antimycobacterial treatment. The definition of treatment success was without re-emergence of multiple positive cultures or persistence of positive cultures with the causative MABC from respiratory samples after ≥12 months of antimycobacterial treatment [22].

### 2.7. Statistical Analyses

Statistical analyses were performed in Stata version 14 (StataCorp LP: College Station, TX, USA). Group comparisons for continuous data were analyzed by Mann–Whitney *U*-test, and group comparisons of proportions by Pearson’s chi-squared test or Fisher’s exact test, where appropriate. The risk factors such as age, gender, acid-fast smear, underlying conditions, radiographical scoring, antimicrobial agents, and surgical resection associated with microbiological and therapeutic outcomes were fitted in the multivariate logistic regression model.

## 3. Results

### 3.1. Patient Enrollment

During the study period, 73 patients who met the 2020 ATS/IDSA criteria for MABC-LD were enrolled. Of them, 26 were from NTUH, 3 from the NTUH YL branch, 6 from TVGH, 13 from CGMH, 7 from EDH, and 18 from FEMH. Among the 73 patients, 2 patients were lost follow-up within 6 months. A total of 71 patients were included for the final assessment of treatment outcome. Of these, 45 had received subspecies identification for their MABC subspecies and 34 had AST (Figure 1).

### 3.2. Patient Characteristics

The baseline characteristics of the 71 patients are summarized in Table 1. The mean age was 63.6 years, and forty-seven (66.2%) were female. Most patients were non-smokers (87.3%). Twelve patients had prior tuberculosis, and most patients were acid-fast positive (83.1%). The mean radiologic score was 5.94 ± 3.11 and higher in subgroups of microbiology failure and treatment failure than their counterpart subgroups.

Thirty-eight MABC species were *M. abscessus* subsp. *abscessus* (MAB), seven were *M. abscessus* subsp. *massiliense* (MMA), none was *M. abscessus* subsp. *bolletii* (MBO), and the other 26 MABC strains were unidentified.

### 3.3. Antimicrobial Susceptibility Testing

Thirty-four MABC strains were tested for in vitro susceptibility. The susceptibility patterns of 11 antimicrobial agents are summarized in Table 2. TGC showed probable excellent minimal inhibitory concentration (MIC) activity against MABC [16]. Ninety-seven percent of TGC MIC were ≤1 mg/L for MABC. The MIC_50_ and MIC_90_ of TGC for MABC were 0.25 mg/L and 0.5 mg/L, respectively. AMI also demonstrated great MIC activity with 94.1% (32/34) susceptibility against MABC. The MIC_50_ and MIC_90_ of AMI for MABC were 16 mg/L and 16 mg/L. CLA at the ERT also showed good MIC with 91.2% (31/34) susceptibility. However, the LRT susceptibility of CLA dropped to 38.2% (12/34). The resistance rates of MABC were high for FQs, with 76.5% (26/34) resistant to CIP and 82.4% (28/34) resistant to MXF. Moreover, we found the susceptibility rates of DOX and MIN were poor against MABC. The susceptibility rate of DOX was 2.9% (1/34), and that of MIN was 11.8% (4/34) against MABC. IMI demonstrated 47% intermediate MIC against MABC.

### 3.4. Treatment Prognosis and Modalities

Table 3 presents the outcomes of different treatment modalities. The overall prognosis for MABC-LD was poor. Twenty-eight patients (39%) achieved microbiology success and 43 patients (61%) had microbiology failure. Twenty-seven patients (38%) achieved treatment success, and 44 patients (62%) had treatment failure (Table 3). Among the 71 patients, only 14 patients received parenteral antibiotics for more than 4 weeks as companion drugs. The rest of the 57 patients received parenteral antibiotics for less than 4 weeks or did not receive parenteral antibiotics as companion drugs. For the patients who received parenteral antibiotics for more than 4 weeks, 11 patients received AMI treatments, 8 patients received IMI, 2 patients received both IMI plus AMI, 5 patients received FQ plus AMI and 10 patients received TGC plus AMI. The average duration for parenteral AMI, IMI, and TGC were 178, 45, and 359.7 days, respectively.

Of the 71 patients, 11 did not receive macrolides; 7 of them failed in microbiology outcome (16.3% of total microbiology failures, 7/43) and seven patients failed in treatment outcome (15.9% of total treatment failure, 7/44). Thirty-eight patients received CLA treatment. However, there were no significant differences in either microbiology success or treatment success compared with those without a macrolide-containing regimen (*p* = 0.63 and 0.47, respectively). Similarly, there were no microbiology or treatment outcome differences in the azithromycin (AZI) treatment group (*p* = 0.56 and 0.43, respectively).

In all, 34 strains were tested for antimicrobial susceptibility. Inducible macrolide resistance was observed in 22 strains (65%) (1 [14%] in MMA and 21 [75%] in MAB, *p* = 0.014). The outcome for patients with delayed macrolide resistance had no significant differences in either microbiology or treatment success (59.1% vs. 40.1%, *p* = 0.91; 40.1% vs. 59.1%, *p* = 0.91). Twelve patients were macrolide susceptible. Fifty-seven patients had treatment regimens without four weeks of parenteral antimicrobial agents. Treatment regimens without parenteral antimicrobial agents achieved less microbiology success and less treatment success, though the difference did not meet statistical significance (64.9 vs. 35.1%, *p* = 0.13; 66.7 vs. 33.3%, *p* = 0.37). For the AMI treated group, the microbiology and treatment outcome seemed better; but the difference did not achieve statistical significance (44.4 vs. 55.6%, *p* = 0.10; 44.4 vs. 55.6%, *p* = 0.37).

Eight patients received IMI intravenous treatments. Only one patient achieved microbiology and treatment success. In the IMI-treated group, the mean radiologic score was 8.63 ± 1.8, and the microbiology and treatment failure rates were both 87.5%. Significantly greater success was observed among patients treated with the combination of TGC and AMI (none use TGC alone or with other parenteral agents). Ten patients received TGC plus AMI intravenously. The mean radiologic score for patients treated with TGC was 3.1 ± 2.74. In comparison with those of other regimens, the microbiology success was 80 vs. 20% (*p* = 0.005) and the treatment success was 80 vs. 20% (*p* = 0.02, Figure 2).

### 3.5. Effect of Antimicrobial Agents on Microbiology and Treatment Outcomes

Multivariate logistic regression analysis adjusted for age, gender, acid-fast smear, comorbidities, radiologic findings, and surgical resection revealed the combination of intravenous TGC and AMI was associated with better microbiology success (adjusted OR 17.724, *p* = 0.03) and treatment success (adjusted OR 14.085, *p* = 0.04), (Table 4). Regimens containing macrolide, AMI alone, or FQ did not show superior outcomes in microbiology or treatment outcomes after adjustments in multivariate logistic regression analysis. Treatment regimens containing IMI, numerically, seemed to have less microbiology success (adjusted OR 0.193, 95% CI 0.019–2.017) and treatment success (adjusted OR 0.169, 95% CI 0.812–4.831). However, the differences were not statistically significant.

## 4. Discussion

In the present study, TGC and AMI had better in vitro sensitivity for MABC, as in the previous report [23]. However, the role of two new agents, TGC combined with AMI, has rarely been studied in MABC-LD and even more rarely by direct comparison with other regimens in the same study. We reported that >1 month of a regimen containing TGC and AMI was correlated with significantly higher microbiology cure and treatment success than regimens without them. In multivariate analysis, the TGC-containing regimens demonstrated more treatment failure, with an adjusted OR of 14.085 (95% CI 1.103–166.667) and a better microbiology cure rate (adjusted OR: 17.724, 95% CI 1.227–267.206).

In fact, the overall treatment response for MABC-LD was as low as 41.2% [9], possibly due to the lack of effective oral antibiotics and poor understanding of and compliance with the intravenous regimen. For in vitro drug susceptibility, only intravenous TGC and AMI had susceptibility of more than 90%, and IMI had a none susceptible rate of 73.5% (Table 2). Other agents, including intravenous and oral forms, had less than 50% sensitivity for MABC. Notably, the drug susceptibility of TGC for MABC was compatible with those of previous studies [16,17], but few studies have focused on the effect of TGC on MABC-LD. Together with the increased treatment success resulting from the regimen containing TGC plus AMI in the present study, we not only echo the treatment recommendation [12] of an intensive phase of MABC-LD treatment containing intravenous agents, but also suggest a regimen of >1 month based on TGC plus AMI as the first-line medications.

However, the patients in our study who received >1 month of intravenous agents, which might be compatible with “guideline-based therapy”, accounted for only 19.7% of the total sample. That figure might be lower than anticipated but compatible with the 11–31% in a previous NTM-NET investigation [13]. The possible explanations include indolent disease course, low awareness of the disease outcome, poor adherence to intravenous antimycobacterial agents, and high worry about the drug’s side effects among both patients and physicians [13,24]. By studying the treatment effect and the importance of intravenous agents for MABC-LD, we can emphasize the correct regimen and promote the treatment in the future.

Although the treatment response of TGC plus AMI was better than those of other regimens, no treatment benefits of IMI or AMI were found in the present study. With regard to IMI, previous individual data meta-analyses showed that intravenous IMI had a treatment benefit for MABC-LD [14,25]. In addition, intravenous AMI has been recommended for the treatment of MABC-LD [14,26]. By contrast, our study did not find treatment benefits for IMI and AMI, possibly due to non-standardized treatment protocol. IMI and AMI were mostly used as salvage therapy, and the disease statuses of patients receiving them were usually more severe, with high radiographical scores (8.63 ± 1.8 for those using IMI). By contrast, the patients who received TGC plus AMI began that regimen from the initial treatment. Therefore, the treatment effects of other intravenous agents, such as IMI, need to be studied in a future prospective study instead of considering the negative effects of a regimen.

For macrolides, a commonly used medication for MABC-LD, different macrolides had similar effects on MABC-LD treatment in this study, although previous studies revealed that AZI seemed better than CLA for MAB lung disease [14,25]. This might be because if there is no clinical response, physicians in real-world practice added or adjusted medications to the guidelines instead of macrolide resistance or subspecies, which are not available in real-time. It is difficult to judge the responsiveness that comes from one drug. In addition, we did not test drug susceptibility and subspecies for all patients. Among the test, only 38.2% of the strains still showed delayed macrolide susceptibility, relatively lower than the percentage in a previous review [10]. The small number might also underestimate the effect. Therefore, we cannot monitor differences in treatment success by different macrolides, resistance, and subspecies.

The strength of the present study is its presentation of a real-world case wherein the combination of TGC and AMI seemed superior to other regimens for MABC-LD, together with their high in vitro susceptibility. There are several limitations of the study. First, the sample was small; therefore, the treatment effects of some medications might have been underestimated. In addition, the retrospective study design with a non-standardized protocol of treatment and follow-up might have biased some results on disease severity, treatment timing, and sputum monitoring. In addition, therapeutic drug monitoring was not a routine practice. No data on the pharmacological therapeutic monitoring of drugs were presented. The effect of therapeutic concentration on microbiology and treatment outcomes was not assessed. Because some strains were not available retrospectively, subspecies identification and drug susceptibility were not performed for all. The effect of subspecies and resistance might be underestimated. Third, because the present study was conducted in Taiwan, further generalization should be performed after validation, for NTM and its resistance are distributed differently.

## 5. Conclusions

In conclusion, MABC-LD is refractory to many anti-MABC drugs, leading to a low treatment success rate. Although a contemporary treatment recommendation has been issued, compliance in the initial intensive phase with intravenous agents is very low (<20%), and this low compliance might be directly correlated with treatment failure. We found that an initial regimen of intravenous TGC plus AMI for >1 month was associated with a high success rate of MABC-LD treatment by a high in vitro susceptibility rate. Further application of the regimen is expected to improve clinical outcomes.

## Figures and Tables

**Figure 1 antibiotics-11-00571-f001:**
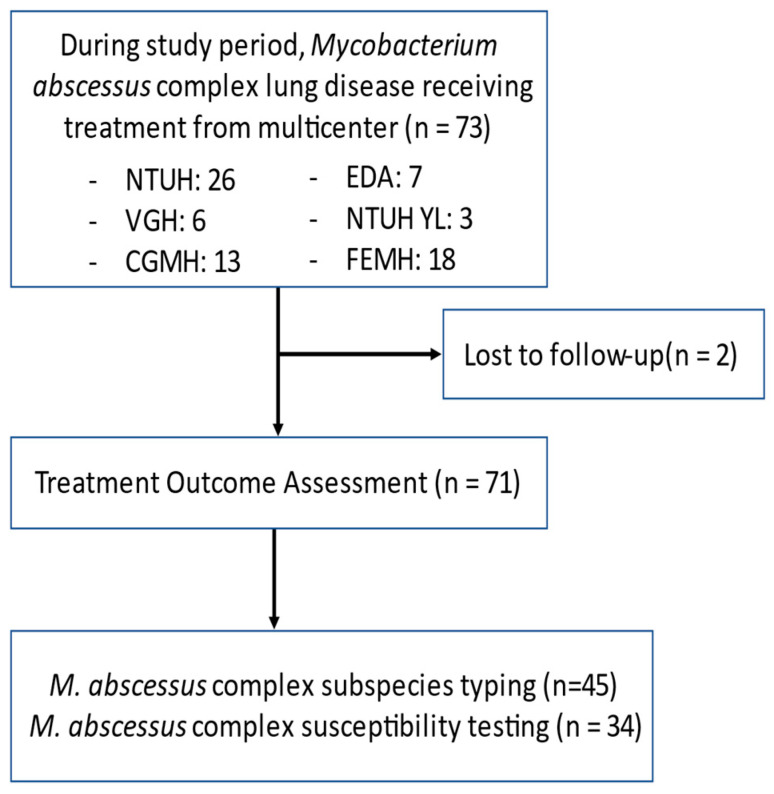
Flow diagram of participant enrollment. Abbreviations: CGMH, Chang Gung Memorial Hospital, Linko Branch; EDH, E-Da Hospital; Far Eastern Memorial Hospital; NTUH, National Taiwan University Hospital; NTUH YL, NTUH YunLin branch; and TVGH, Taipei Veterans General Hospital.

**Figure 2 antibiotics-11-00571-f002:**
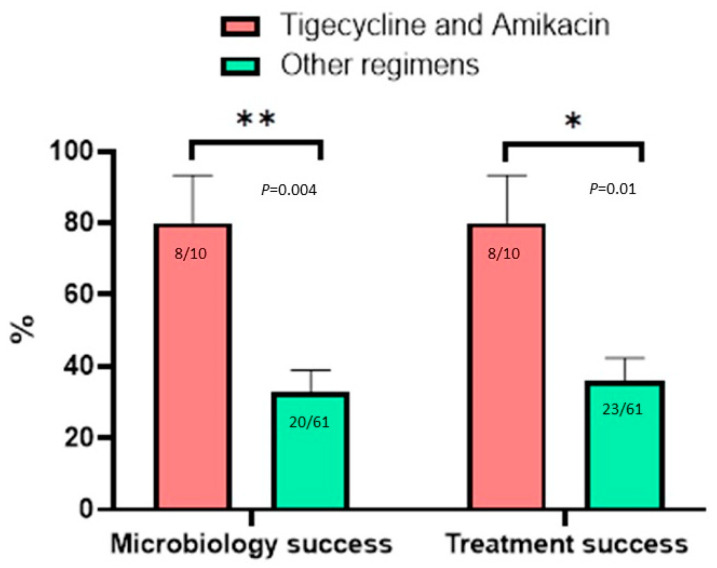
The outcomes of the patients with *Mycobacterium abscessus* complex lung disease according to treatment regimens with or without tigecycline and amikacin. ** *p* = 0.004, * *p* = 0.01.

**Table 1 antibiotics-11-00571-t001:** Demographics and clinical characteristics of patients with *Mycobacterium abscessus* complex lung disease.

	Total (N = 71),	Microbiology Failure (N = 43)	Microbiology Cure (N = 28)	*p* Value	Treatment Failure (N = 44)	Treatment Success (N = 27)	*p* Value
Age, years	63.6 ± 13.07	64.3 ± 13.6	62.5 ± 12.2	0.68	64.1 ± 13.6	62.9 ± 12.4	0.69
Gender, female	47 (66.2)	30 (69.8)	17 (60.7)	0.43	30 (68.2)	17 (62.9)	0.65
**Body weight, Kg**	50.4 ± 10.6	50.8 ± 10.9	49.8 ± 0.2	0.27	50.9 ± 10.9	49.8 ± 10.2	0.27
**Non-smoker**	62 (87.3)	38 (88.4)	24 (85.7)	0.74	39 (88.6)	23 (85.2)	0.67
**Underlying disease**					8 (18.2)	4 (14.8)	0.71
Previous pulmonary tuberculosis	12 (16.9)	8 (18.6)	4 (14.3)	0.63	5 (11.4)	1 (3.7)	0.26
DM	6 (8.5)	5 (11.6)	1 (3.6)	0.23	1 (2.3)	1 (3.7)	0.69
ESRD	2 (2.8)	1 (2.3)	1 (3.6)	0.75	7 (15.9)	4 (14.8)	0.5
Malignancy	11 (15.5)	7 (16.3)	4 (14.3)	0.82	4 (9.1)	4 (14.8)	0.95
Rheumatic disorder	8 (11.3)	6 (14)	2 (7.1)	0.37	3 (6.8)	2 (7.4)	0.92
Asthma	5 (7.0)	3 (7)	2 (7.1)	0.97	8 (18.2)	5 (18.5)	0.56
COPD	13 (18.3)	7 (16.3)	6 (21.4)	0.63	36 (81.8)	23 (85.2)	0.29
**Sputum acid fast smear, positive**	59 (83.1)	36 (83.7)	23 (82.1)	0.86			
**Radiographic pattern**					13 (29.5)	5 (18.5)	0.30
Fibrocavitation	18 (25.4)	13 (30.2)	5 (17.9)	0.24	36 (81.8)	23 (85.2)	0.35
Nodular bronchiectasis	59 (83.1)	35 (81.4)	24 (85.7)	0.63	64.1 ± 13.6	62.9 ± 12.4	0.69
Radiographic score	5.94 ± 3.11	6.86 ± 3.42	4.46 ± 2.14	0.001	7.21 ± 3.22	4.64 ± 2.41	<0.001
**Surgical resection**	6 (8.5)	3 (7)	3 (10.7)	0.58	3 (6.8)	3(11.1)	0.86
***Mycobacterium* species**							
*M. abscessus* subsp., unclassified	26 (36.6)	17 (39.5)	9 (32.1)	0.66	18 (40.9)	8 (29.6)	0.78
*M. abscessus* subsp. *massiliense*	7 (9.9)	5 (11.6)	2 (7.1)	0.53	5 (11.3)	2 (7.4)	0.23
*M. abscessus* subsp. *abscessus*	38 (53.5)	21 (48.8)	17 (60.7)	0.59	21 (47.7)	17 (62.9)	0.65

Data are shown as no. (%) or mean ± standard deviation. Abbreviations: DM, diabetic mellitus; ESRD, end-stage renal disease; COPD, chronic obstructive pulmonary disease.

**Table 2 antibiotics-11-00571-t002:** In vitro antimicrobial susceptibility of 34 *Mycobacterium abscessus* complex isolates.

	MIC * (µg/mL)			MIC_50_ (µg/mL)	MIC_90_ (µg/mL)	Range (µg/mL)
	Susceptible	Intermediate	Resistant			
**TMP-SMX**	**≤2**		≥4			
	17 (50)	0	17(50)	4	16	0.25–16
**Ciprofloxacin**	≤1	2	≥4			
	4 (11.8)	4 (11.8)	26 (76.5)	4	8	0.25–8
**Moxifloxacin**	≤1	2	≥4			
	3 (8.82)	3 (8.82)	28 (82.4)	8	16	0.5–16
**Cefoxitin**	≤16	32	≥64			
	4 (11.8)	14 (41.2)	16 (47.1)	32	128	4–256
**Amikacin**	≤16	32	≥64			
	32 (94.1)	1 (2.9)	1 (2.9)	16	16	4–64
**Doxycycline**	≤1	2	≥4			
	1 (2.9)	3 (8.8)	30 (88.2)	16	32	0.5–32
**Clarithromycin ERT ****	≤4	8	≥16			
	31 (91.2)	0	3 (8.8)	0.25	1	0.125–32
**Clarithromycin LRT** *******	≤4	8	≥16			
	12 (35.3)	0	22 (64.7)	16	32	0.125–32
**Imipenem**	≤8	16	≥32			
	9 (26.5)	16 (47)	9 (26.5)	16	32	8–128
**Minocycline**	≤4		≥8			
	4 (11.8)	0	30 (88.2)	8	16	1–16
**Linezolid**	≤8	16	≥32			
	6 (17.6)	7 (20.6)	21 (61.8)	32	64	2–64
**Tigecycline**	≤1		≥2			
	33 (97)	0	1 (3)	0.25	0.5	0.125–2

TMP-SMX = trimethoprim-sulfamethoxazole. * Minimal inhibitory concentration cutoffs were adopted from the Clinical and Laboratory Standards Institute [20] except tigecycline [16]. Data were shown as no. (%). ** ERT: early reading time (usually reading on the 3rd to 5th day when growth is optimal).*** LRT: late reading time (reading on the 14th day).

**Table 3 antibiotics-11-00571-t003:** Comparing treatment regimen and microbiology in different treatment outcome.

Treatment Modality	Total (N = 71)	Microbiology Failure (N = 43)	Microbiology Success (N = 28)	*p* Value	Treatment Failure (N = 44)	Treatment Success (N = 27)	*p* Value
**Macrolide use**							
Clarithromycin	38	24 (63.2)	14 (36.8)	0.63	25 (65.8)	13 (34.2)	0.47
Azithromycin	30	17 (56.7)	13 (43.3)	0.56	17 (56.6)	13 (43.3)	0.43
Non-macrolide use	11	7 (63.6)	4 (36.4)	0.82	7 (63.6)	4 (36.4)	0.92
**Resistance**							
Delayed macrolide resistance *	22	13 (59.1)	9 (40.1)	0.91	9 (40.1)	13 (59.1)	0.91
Macrolide susceptible	12	7(58.3)	5(41.7)	0.86	7 (58.3)	5 (41.7)	0.71
**Parenteral drug**							
Parenteral drug use <4 weeks	57	37 (64.9)	20 (35.1)	0.13	38 (66.7)	19 (33.3)	0.37
Parenteral drug use ≥4 weeks	14	6 (42.9)	8 (57.1)	0.13	6 (42.9)	8 (57.1)	0.37
Amikacin	18	8 (44.4)	10 (55.6)	0.1	8 (44.4)	10 (55.6)	0.37
Imipenem	8	7 (87.5)	1 (12.5)	0.09	7 (87.5)	1 (12.5)	0.04
Fluoroquinolone	32	21 (65.6)	11 (34.4)	0.43	21 (65.6)	11 (34.3)	0.95
Imipenem and amikacin	2	2 (100)	0	0.52	2 (100)	0	0.49
Fluoroquinolone and amikacin	5	4(80)	1(20)	0.64	4 (80)	1(20)	0.36
Tigecycline and amikacin	10	2 (20)	8 (80)	0.005	2 (20)	8 (80)	0.02

Data were no. (%). * 36 patients had delayed macrolide susceptibility test data, and a total of 22 patients with delayed macrolide resistance.

**Table 4 antibiotics-11-00571-t004:** Multivariable analysis for treatment outcomes according to Individual Antibiotics.

Antibiotics	Microbiology Success			Treatment Success		
	Adjusted OR	95% CI	*p*-Value	Adjusted OR	95% CI	*p*-Value
**Macrolide**	1.097	0.227–5.292	0.9	0.719	0.151–3.425	0.67
**Amikacin**	0.771	0.105–5.672	0.79	0.656	0.812–4.831	0.67
**Imipenem**	0.193	0.019–2.017	0.17	0.169	0.330–1.808	0.14
**Fluoroquinolone**	1.688	0.463–6.155	0.14	2.487	0.698–8.850	0.16
**Tigecycline and amikacin**	17.724	1.227–267.206	0.03	14.085	1.103–166.667	0.04

Abbreviations: OR, odds ratio; CI, confidence interval. Adjusted for age, gender, acid-fast smear, underline conditions, radiologic findings, and surgical resection.

## Data Availability

Not Applicable.

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
