# Peer review of "Treatment Outcome in Patients with Mycobacterium abscessus Complex Lung Disease: The Impact of Tigecycline and Amikacin"

_antibiotics, 2022, doi:10.3390/antibiotics11050571_

Round 1

Reviewer 1 Report

This is an analysis of the impact of tigecycline and amikacin therapy for the treatment of patients with Mycobacterium abscessus complex lung disease. Tigercycline, a new antibiotic for MABC, has not yet been well studied for MABC-LD. This study was conducted to investigate the effect of this treatment. However, multiple details must be provided to justify a publication. The criteria chosen for the evaluations of microbiological therapeutic success seem to merge and therefore need to be reviewed. Page 4 : study population : Why didn't you randomize your patients into two groups? Page 4 : Administration and assessment of effective treatment regimens : Explain how the treatments were assigned. Page 5 : Assessment of outcomes : Outcome evaluation was categorized as microbiological cure and treatment success. The definitions of microbiological cures and treatment success are not exactly those defined in the international consensus. In this publication the results are similar for microbiological cure and therapeutic success because the definitions are practically the same for these two parameters. Page 5: Results analyses No data on the pharmacological therapeutic monitoring of drugs are presented. However, it would be very interesting to bring the results back to the concentrations measured in your patients. Page 5: Results analyses A subgroup analysis is missing for strains with known clarithromycin status. This is essential for the analysis. Page 5: Results analyses The results are completely similar in microbiological cure and treatment success. This is due to the definition of these two parameters which is not the correct one. Review the results with the correct definition.

Author Response

Dear reviewer 1:

Thanks for your informative comments and helpful suggestions. Your opinion provides constructive improvements for our article. Here is some of my point-to-point response for your comments:

  1. Page4: Why don’t you randomize patients into two groups and how the treatments were assigned?

Reply: This is a retrospective study conducted during 2009 to 2018. We reviewed the medical charts and bacterial analysis retrospectively. This is a quasi-experimental study; therefore, the patients were not randomized into tigecycline-containing or non-tigecycline-containing group. The patients were enrolled from 6 different medical centers in different areas. Treatment protocols were assigned by patients’ physician in charge individually. However, the antibiotics managements were mostly following the old version ATS guidelines in 2007 including macrolides, fluoroquinolones, aminoglycosides and tigecycline. The treatment modality was summarized in Table 3.

  1. Page 5: Assessment of outcomes is not current consensus.

Reply: Thank you for the comments. We adopted the definition of treatment outcomes by an NTM-NET consensus statement (van Ingen J, Aksamit T, Andrejak C, Bottger EC, Cambau E, Daley CL, et al. Treatment outcome definitions in nontuberculous mycobacterial pulmonary disease: an NTM-NET consensus statement. Eur Respir J. 2018;51). For each definition, the one that received most votes was selected for assessment in our article.

  1. Page 5: No data on the pharmacological therapeutic monitoring of drugs are presented.

Reply: Indeed, the therapeutic drug monitor was lacking in our research. Most of the treatments were followed by 2007 ATS guideline suggestion. The old guideline did not suggest therapeutic monitoring as routine practice. It is a great advice for us to conduct therapeutic drug concentration and analyze the results with outcomes.  We added this point into the limitation of the Discussion section.( Discussion, line 321-323)

  1. Page 5: Results analyses A subgroup analysis is missing for strains with known clarithromycin status. This is essential for the analysis。

Reply: In our study, we collected 71 patients fulfilling pulmonary NTM infection. These patients coming from 6 different medical center. Not all hospital performed subspecies identification and antimicrobial susceptibility testing. Of these, 45 had received subspecies identification and 34 had antimicrobial susceptibility testing (Figure 1). Therefore, the subgroup analysis was analyzed in the 34 patients who had clarithromycin susceptibility testing. Owing to small case numbers, resistance did not shown the influence for the outcome. We added this point into the limitation of the Discussion section.( Discussion, line 317-318)

Reviewer 2 Report

1. The subject of the publication is important and interesting.

2. However, I have some comments that need to be supplemented in the text of the article:

  • the authors did not explain why they use the data of statistical tests (e.g. why is there a Mann Whitney U test instead of a parametric t-Student test?). I am asking for your opinion on this in the text.
  • significance levels are described in the analysis of the results (similar to the tables). However, I miss a clear response to this: is the lack of significant differences (or significant differences elsewhere) something positive or negative in relation to the substantive issues?
  • the article lacks (apart from the last paragraph in the discussion) a summary of the research carried out and writing (e.g. in points) of cognitive conclusions; it would also be worth writing something about the prospects for further research.

Author Response

Dear reviewer 2:

Thanks for your informative comments and helpful suggestions. Your opinion provides constructive improvements for our article. Here is some of my reply for your comments:

  1. the authors did not explain why they use the data of statistical tests (e.g. why is there a Mann Whitney U test instead of a parametric t-Student test?

Reply: During our study period, a total of 71 patients were included for final assessment of treatment outcome. Of these, 45 had received subspecies identification and 34 had antibiotics susceptibility test. The patient numbers were not enough for normal distribution to carry out parametric Student t test.  Therefore, we use non-parametric statistical test instead. Thank you for your comment.

Is the lack of significant differences (or significant differences elsewhere) something positive or negative in relation to the substantive issues?

Reply: Thank you. Macrolides possess potent activity against M. abscessus as well as immunomodulatory effects. Even for patients with inducible macrolide resistant M.abscessus, a macrolide containing regimen is still advised for treatments. However, In our study, there were no significant differences in either microbiology success or treatment success comparing regimens with or without a macrolide containing. That means, our p-value is unexpectedly over the arbitrary threshold for ‘significance’ of p=0.05. Though the results were so called “negative”, we still reported the results. We still believe our research possess a positive impact toward clinical practice.

  1. the article lacks (apart from the last paragraph in the discussion) a summary of the research carried out and writing (e.g. in points) of cognitive conclusions; it would also be worth writing something about the prospects for further research.

Reply: Thanks for your kindly advise. It is of great helpful suggestions. We would summarize our results and add a brief conclusion in our revised version. (See conclusion section)

Reviewer 3 Report

This kind of retrospective study will be important to lead the best (better) regimen for the treatment of MABBC-LD patients. The number of data in this manuscript is still low, but the suggested regimen may be helpful for the MABBC-LD patients. The following points are considered in the revised manuscript;

1) p.4, line 6 : YunLin (YL)

2) p.6 : The effective number of patients is 71, but the number of MAB subspecies typing is 45, not 71. Explain the reason. Also, the number of MAB susceptibility testing is 34, not 45. Explain the reason.

3) p.6, line 7 in "Patient characteristics" : none was M. abscessus ..

4) p.8, Table 2 : It will be interesting if difference of antimicrobial susceptibility among the MAB subspecies. No difference among them?

5) p.9,  Delayed macrolide resistance in Table 3 :The number of Microbiology success is 9, while he number of treatment success increased 13. Is it curious? Please explain.

6) p.11, line 12 in "Discussion" : cannot follow the numbers (more than 90%, and 73.5% in Table2. Please modify/revise them.

7) p.11, line 21 in "Discussion" : cannot follow the number (19.7%). Please explain.

Author Response

Dear reviewer 3:

Thanks for your informative comments and helpful suggestions. Your opinion provides constructive improvements for our article. Here is some of my reply for your comments:

1) p.4, line 6 : YunLin (YL)

Reply: the abbreviation is corrected now.

2) p.6 : The effective number of patients is 71, but the number of MAB subspecies typing is 45, not 71. Explain the reason. Also, the number of MAB susceptibility testing is 34, not 45. Explain the reason.

Reply: During our study period, a total of 71 patients were included for final assessment of treatment outcome. Of these, 45 had received subspecies identification and 34 had antibiotics susceptibility test (figure 1). Our patient enrollments coming from 6 different medical centers in different area. Not all hospitals performed subspecies identification and antimicrobial susceptibility testing as their routine practice. That’s why the numbers of species subtyping and drug susceptibility is not 71. We added this point into the limitation of the Discussion section.( Discussion, line 317-318)

3) p.6, line 7 in "Patient characteristics" : none was M. abscessus ..

Reply: The grammar error is corrected now.

4) p.8, Table 2 : It will be interesting if difference of antimicrobial susceptibility among the MAB subspecies. No difference among them?

Reply: We have 34 mycobacterium abscessus complex isolates had antimicrobial susceptibility testing. Of them, only 5 isolates were marsilliense. Though all the 5 marsilliense were clarithromycin susceptible, the difference did not achieve statistical significant in our data.

5) p.9,  Delayed macrolide resistance in Table 3 :The number of Microbiology success is 9, while he number of treatment success increased 13. Is it curious? Please explain.

Reply: We adopted the definition of treatment outcomes by an NTM-NET consensus statement (van Ingen J, Aksamit T, Andrejak C, Bottger EC, Cambau E, Daley CL, et al. Treatment outcome definitions in nontuberculous mycobacterial pulmonary disease: an NTM-NET consensus statement. Eur Respir J. 2018;51). In this consensus, The definition of microbiology cure was finding multiple consecutive negative cultures for causative MABC from respiratory samples after culture conversion and until the end of antimycobacterial treatment. The definition of treatment success was without re-emergence of multiple positive cultures or persistence of positive cultures with the causative MABC from respiratory samples after ⩾ 12 months of antimycobacterial treatment. By the definition, some patients did not achieve culture conversion at the end of treatments. But due to improved clinical symptoms, the treating physician ceased their antibiotics treatments and found they achieve culture conversion after ⩾ 12 months follow-ups. Therefore, the number of treatment success increased.

6) p.11, line 12 in "Discussion" : cannot follow the numbers (more than 90%, and 73.5% in Table2. Please modify/revise them.

Reply: Thanks for pointing out the typing error. The IMI had a none susceptible rate of 73.5%.

            The typing error is corrected now.

7) p.11, line 21 in "Discussion" : cannot follow the number (19.7%). Please explain.

Reply: In 71 patients, only 14 patients received parenteral antibiotics more than 4 weeks. The percentage is 19.7% (14/71) of the total sample.

Round 2

Reviewer 1 Report

Dear authors,   Thank you for taking my remarks and comments into account.   Kind regards